# On-Line Fluorescence Microscopy for Identification and Imaging of Apoptotic Cell with Synchrotron-Based Soft X-ray Tomography

**DOI:** 10.3390/mi14020326

**Published:** 2023-01-27

**Authors:** Chao Zhang, Zhao Wu, Zheng Dang, Lijiao Tian, Yong Guan, Gang Liu, Yangchao Tian

**Affiliations:** National Synchrotron Radiation Laboratory, University of Science and Technology of China, Hefei 230029, China

**Keywords:** on-line fluorescence microscopy, soft X-ray tomography, apoptotic cells

## Abstract

Synchrotron-based soft X-ray tomography (SXT), providing three-dimensional morphology and quantitative distribution of linear absorption coefficient (LAC) of the imaged objects, is widely used in many fields to obtain ultra-structure images, especially in cellular imaging. Off-line fluorescence microscopies (FMs) are combined to identify the type of organelles and status of cells. However, deformation and displacement usually occur during the transfer and loading process, which decreases the precision of two-modal images’ registration. In this paper, we report on an on-line FM, at the SXT station (BL07W) of the National Synchrotron Radiation Laboratory (NSRL), which avoids deformation and displacement. Therefore, researchers can easily find the sample and take the useful data without tedious post-processing. Combining SXT with on-line FM, we achieved the identification and high-resolution imaging of an apoptotic cell. The experiments revealed that the LAC of the nucleus of the apoptotic cell was larger than that of a normal cell, which could be explained by nucleus pyknosis of the apoptotic cell.

## 1. Introduction

Over the years, synchrotron-based soft X-ray tomography (SXT) has been demonstrated to be a powerful microscope in observing the three-dimensional (3D) structure of objects in biological [1,2,3] and materials science [4,5] research. Since soft X-rays cover the *K*-shell absorption edges of carbon (284 eV) and oxygen (543 eV), cryogenic SXT (cryo-SXT) allows high contrast imaging of cells in their near natural state. SXT provides 3D morphology and quantitative distribution of the linear absorption coefficient (LAC). Off-line fluorescence microscopies (FMs) are usually combined to identify types of organelles and cells [6,7,8]. With correlative off-line FM and cryo-SXT, researchers can easily obtain the state/function information and ultrastructure of the imaged cells [8,9,10,11,12,13,14,15,16,17], but deformation and displacement usually occur due to its being ex-situ. 

The off-line correlative method is usually performed with fluorescence imaging and SXT in sequence. Samples need to be transferred between different imaging facilities, and the samples selected in off-line FM may be damaged or shifted during the transfer and loading process. More importantly, the deformation and displacement impose difficulty in image registration. Therefore, on-line FM was proposed to solve these problems. With on-line FM, researchers can easily find the sample buried in an ice layer and take the useful data without tedious post-processing. Most stations of synchrotron-based SXT around the world, such as B24@Diamond [18] and MISTRAL@ALBA [19], utilize an LED light source and filter module to build an on-line FM based on a built-in visible light microscope (VLM) system. This has been employed to find suitable samples with fluorescent markers during soft X-ray imaging. However, the usage of on-line FM revealing the status or functions of cells has not been reported on. 

In order to attain the structural and functional information of a cell concurrently, we built an on-line FM with long working distance objectives to construct an on-line correlative imaging platform. Laser and sCMOS were, respectively, adopted as excitation source and detector to capture high-quality fluorescence images, benefiting from the high-intensity illumination and high-sensitivity detector. Then, identification of the status of cells (normal, apoptotic or necrotic) could be achieved. With SXT, we obtained their 3D structures and the LAC of organelles by means of a self-developed iteration algorithm. From the experimental results, we found the LAC of the nucleus of an apoptotic cell was larger than that of a normal cell, which could be explained by nucleus pyknosis of the apoptotic cell. 

## 2. Materials and Methods

### 2.1. Optical Design and Implementation

The optical layout of soft X-ray microscopy (SXM) and on-line FM at BL07W@NSRL are shown in Figure 1. The soft X-ray imaging system consists of a condenser, sample holder, zone plate (ZP) and detector. These components are arranged in the vacuum SXM chamber. The condenser concentrates the incident soft X-rays to the sample with a hollow cone illumination, and the transmitted X-rays are focused by the zone plate to form a magnified image on the detector. The optical axis of the VLM, which is helpful to find suitable samples before performing SXT, is at an angle of 60° to the X-ray optical axis. By introducing a laser and filter module into the VLM, an on-line FM is constructed. Therefore, it could be used to image and find adherent cells stained by fluorescent markers on a cryogenic sample grid.

In order to develop the fluorescence imaging capabilities of the on-line VLM, we designed the excitation unit and fluorescence collection unit at the BL07W@NSRL station. Due to the limited space of the original platform, we placed a new optical table nearby to arrange the optical components of the excitation unit, as shown in Figure 1b.

Figure 1c shows the zoomed optical layout of the excitation unit of the on-line FM. We designed two high-power density semiconductor lasers with wavelengths of 405 nm (OBIS 405LX, Coherent, Saxonburg, USA) and 488 nm (OBIS 488LS, Coherent) as the excitation light, which allowed for super-resolution FM. The system could be upgraded to a photoactivated localization microscopy (PALM) system by replacing the objective lens by one with a larger numerical aperture [20]. Moreover, lasers could provide energy for some chemical reactions in *in situ* study by SXT [21,22]. Two laser beams with diameters of about 0.7 mm to 0.8 mm were combined through the mirror (M1-M3; BB05-E02, Thorlabs, Newton, USA) and dichroic mirror (DM; DMLP425, Thorlabs) and then passed through an acousto–optical tunable filter (AOTF; AOTFnC-400.650, AA Opto Electronic, Orsay, France) for selection of wavelength and power adjustment. The lasers were tripled in diameter by the beam expansion (BE) system and reflected by the mirror M4 (BB05-E02, Thorlabs) onto the sample at SXT stage with a smaller divergence angle (Figure 1c). 

Figure 1d shows the interface of the on-line FM and SXM chamber. Mirror M5 reflected laser from horizontal orientation to vertical orientation and the mirror M6 reflected the beam into the SXM chamber. Then, a focusing lens (FL; AC254-500-A, Thorlabs) with a focal length of 500 mm focused the beam to the back focal plane of the objective lens inside the chamber and the laser was epi-illuminated on the sample. The sCMOS (400D, Dhyana, Fuzhou, China), instead of the VLM CCD, captured higher-quality images. In fluorescence imaging mode, the fluorescence excited by the laser was collected by the sCMOS through the multi-band DM (ZT405/488/561/640rpcv2, Chroma, Vermont, USA), emission filter (EM; ET525/50m, Chroma), tube lens (TL; AC508-200-A, Thorlabs) and mirror M7 (BB05-E02, Thorlabs).

### 2.2. Sample Preparation and Imaging Workflow

Figure 2 displays the experimental workflow. The experiment was started by culturing HeLa cells at 37 °C with 5% CO_2_. The HeLa cell line (#C6330) was bought from a commercialized company, Beyotime [23]. Then, the cells needed to be seeded on the nickel grid for subsequent experiments. The specific method for preparation of sample grids and seeding of cells on the grids were discussed in detail by Okolo, C.A et al. [24].

After the HeLa cells adhered to the grid, the cells were treated with 0.5 μM apoptotic inducer staurosporine for 4 hours. Fluorescent imaging was used to detect apoptosis. Here we chose CellEvent™ Caspase-3/7 Green Detection Reagent (#C10723, Invitrogen) as the fluorescence detection reagent. We added 7.5 μM of the detection reagent to the cells and incubated them at 37 °C with 5% CO_2_ for 30 min, according to the manufacturer’s user guide. This reagent consisted of a four amino acid peptide (DEVD) conjugated to a nucleic acid binding dye. The cell-permeant substrate was intrinsically non-fluorescent, because the DEVD peptide inhibits the ability of the dye to bind to DNA. After activation of caspase-3 or caspase-7 in apoptotic cells, the DEVD peptide was cleaved, enabling the dye to bind to DNA and produced a bright, fluorogenic response with absorption/emission maxima of ~502/530 nm. 

After that, we plunged the grid with adhered cells into liquid nitrogen-cooled liquid ethane. The vitrified grid was transferred to the sample holder of the SXM. The on-line FM and soft X-ray mosaic was used to identify and locate cells in different statuses. First, the sample was rotated to −60° for fluorescence imaging. Then, it was rotated to 0° for soft X-ray imaging of the same region. Following that, the projections of the cells at different angles were captured with SXT. 

### 2.3. Reconstruction Algorithms for Linear Absorption Coefficient

The sample was rotated in a range of ±65° to obtain projections at different angles, and then the 3D structure of the sample could be reconstructed by means of an analytical reconstruction algorithm, such as the filter back projection (FBP), and algebraic iterative reconstruction algorithms, such as the simultaneous iterative reconstruction technique (SIRT) and simultaneous algebraic reconstruction technique (SART). The reconstruction speed of the FBP algorithm is fast, but its reconstruction quality is not satisfactory in the limited-angle SXT. While the iterative algorithm improves the reconstruction quality significantly, the SIRT algorithm has better suppression of artifacts [25,26,27]. Therefore, SIRT was employed in this work to reconstruct the slices. Then the reconstructed slices could be imported into Amira software for segmentation, and the grey value in each segmentation region was counted to obtain a Gaussian distribution. The mean value and variance of the Gaussian distribution equaled the mean and variance of the LAC [28].

## 3. Results and Discussion

Figure 3 displays the images collected by the on-line FM at BL07W@NSRL. Figure 3a shows the bright-field image of the cryogenic grid, with cells adhered in different grid holes. Usually, it is hard to find the cells, due to their low contrast relative to the vitreous ice layer. However, the cells in the red box in Figure 3a were obvious because this grid hole was illuminated simultaneously by a laser of 488 nm, while the cells in the same area became less visible when the laser was turned off, as shown in Figure 3b. Figure 3c shows the fluorescent image of the corresponding region in the red box in Figure 3a, with the bright-field illumination light in the off state. The laser spot size on the sample almost covered a single grid hole, so we could search for cells one hole at a time without worrying about the laser spot quenching the fluorescence of cells in other holes. The results showed that the on-line FM greatly improved the efficiency of finding the sample.

Figure 4 shows the fluorescence images and soft X-ray mosaic of cells in different statuses. Panels (a–d) display the results of cells treated with 0.5 μM staurosporine for 4 hours, while panels (e–f) exhibit the results of normal HeLa cells without treatment. The intense fluorescent signal in Figure 4a,b indicated that the key proteases caspase 3/7 in the apoptotic pathway was successfully activated and also meant that the localized cells were apoptotic cells. However, the fluorescence of the cell in Figure 4c,d was faint, which indicated that it had not undergone apoptosis, although it was also treated with 0.5 μM staurosporine for 4 hours. Therefore, the built on-line FM was capable of identifying and localizing apoptotic cells from those treated with apoptosis inducers. A faint fluorescence signal was also found in normal HeLa cells without treatment, as shown in Figure 4f. These were due to the autofluorescence of the cells. The faint autofluorescence signal could also be easily detected with the help of sCMOS which had its own low light detection capability. Thus, on-line FM could find cells faster and more accurately than the original bright field VLM. 

The soft X-ray mosaics of apoptotic cells, shown in Figure 4a,b, revealed typical features of apoptosis, such as cell shrinkage but with intact cell membranes. While the cells in Figure 4c,d presented cell swelling, disrupted cell membrane and cytoplasm release, which were consistent with the reported characteristics of necrotic cells in published studies [29]. The normal cells in Figure 4e,f were structurally intact, and the nucleolus in the nucleus was clearly visible.

The mean fluorescence intensity of cellular fluorescence in Figure 4 was calculated and shown in Figure 5. Error bars indicated the standard deviations of the fluorescence intensities. The labels a, b, c, d and f correspond to the letter labels in Figure 4. The mean fluorescence intensity of apoptotic cells was 6 to 30 times higher than that of normal and necrotic cells, which could help us to understand the biochemical information of cells. Combined with the structural information of the soft X-ray mosaic, we could easily distinguish normal cells, apoptotic cells and necrotic cells. Therefore, a criterion shown in Table 1 to distinguish cell status on-line could be established.

With the on-line correlative imaging platform and the identification criterion described above, we performed SXT on the nucleus of the cells. Then, slices of the cells shown in Figure 6 could be reconstructed by means of the SIRT algorithm. Panels a-b exhibit the slices of two apoptotic cells in Figure 4a, respectively. Figure 6c,d respectively display the slice of the necrotic cell and the normal cell in Figure 4d,e. It is easy to see that the nuclear membranes of the apoptotic cells were ruptured while the normal cells were intact and the three nucleoli were clearly visible. In addition, by calculating the LAC of the nucleus, we found that the LAC values of apoptotic cells (0.215 ± 0.08 μm^−1^ and 0.248 ± 0.06 μm^−1^ for Figure 6a,b, respectively) were larger than those of normal cells (0.14 ± 0.03 μm^−1^). This indicated that nucleus pyknosis existed in apoptotic cells, which was the most significant feature of apoptosis and caused by chromatin condensation [29]. However, the organelles of necrotic cells were disorganized and were very different from the two types of cells mentioned above, as shown in Figure 6c.

## 4. Conclusions

In this work, an on-line FM was designed and implemented at BL07W@NSRL. In combination with soft X-ray imaging, this correlative imaging platform was successfully used to identify apoptotic cells, necrotic cells and normal cells on-line. Activation of key proteases caspase 3/7 in the apoptotic pathway was successfully confirmed according to the intense fluorescent signals. The difference in overall morphological structure and local 3D structure of cells were revealed by soft X-ray mosaic images and SXT, respectively.

This on-line correlative imaging platform and method would provide insight and assistance in cell imaging. Based on this platform, researchers can easily identify the status and/or function of cells on-line as long as appropriate fluorescent markers are applied. The appropriate cells can then be selected efficiently for SXT and useful data can be obtained without tedious post-processing. The developed platform will greatly help researchers to improve the efficiency and accuracy of their research.

## Figures and Tables

**Figure 1 micromachines-14-00326-f001:**
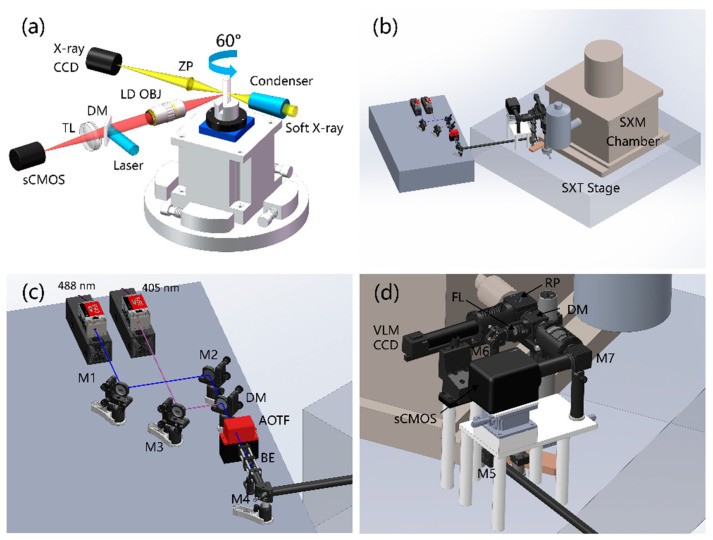
The layout of soft X-ray microscopy and on-line FM. (**a**) Schematic diagram of on-line FM at BL07W@NSRL station; (**b**) implementation diagram of on-line FM; (**c**) zoomed laser excitation unit and (**d**) zoomed fluorescence collection unit in panel (**b**).

**Figure 2 micromachines-14-00326-f002:**
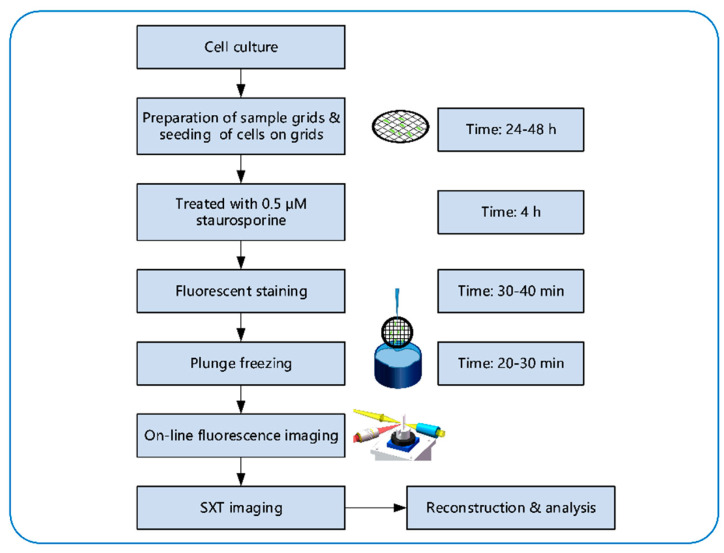
Workflow of the sample preparation and imaging.

**Figure 3 micromachines-14-00326-f003:**
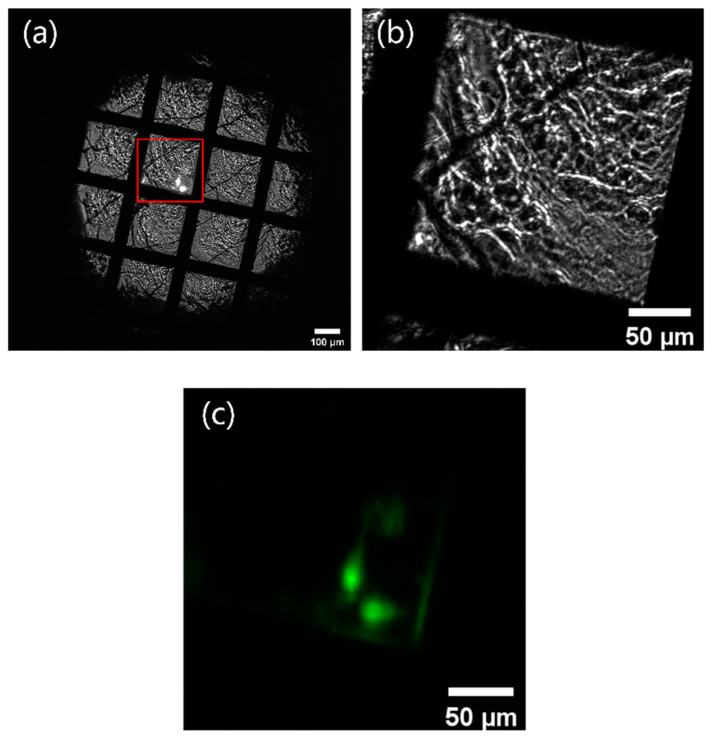
The images collected by on-line FM installed at BL07W@NSRL. (**a**) bright-field images of the cryogenic sample grid captured by sCMOS when the laser was turned on; (**b**) bright-field images of the red box area in Figure 3a when the laser was turned off; (**c**) fluorescent image of the grid in (**b**). All acquired images were taken with the on-line 10× objective.

**Figure 4 micromachines-14-00326-f004:**
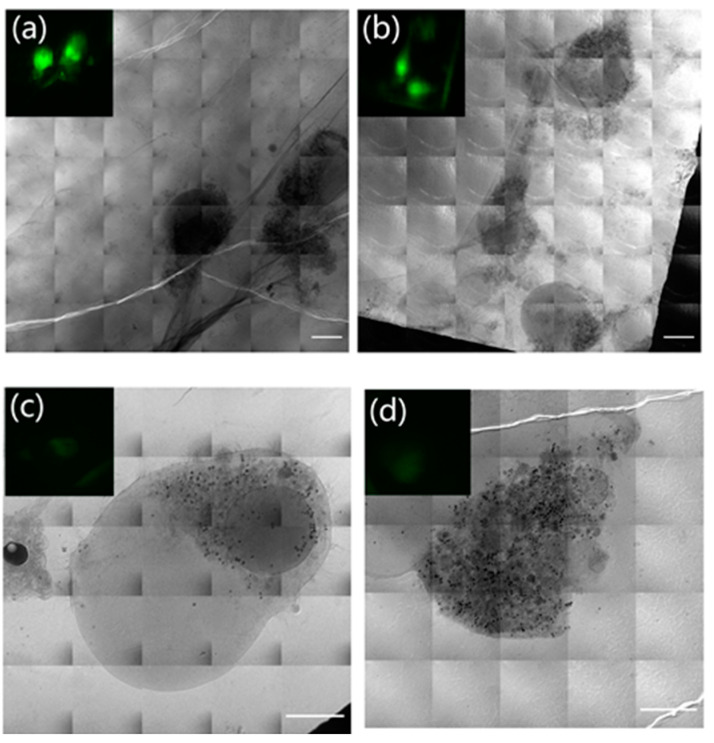
Fluorescent and soft X-ray mosaic images of HeLa cells. (**a**–**d**) HeLa cells treated with 0.5 μM staurosporine for 4 hours. The inset is a fluorescent picture, corresponding to the mosaic. The intense fluorescence in (**a**,**b**) indicates apoptotic cells, while the faint fluorescence in (**c**,**d**) indicates that the cells are not apoptotic; (**e**,**f**) Normal HeLa cells without treatment. Nu indicates nucleolus. Scale bar 10 μm.

**Figure 5 micromachines-14-00326-f005:**
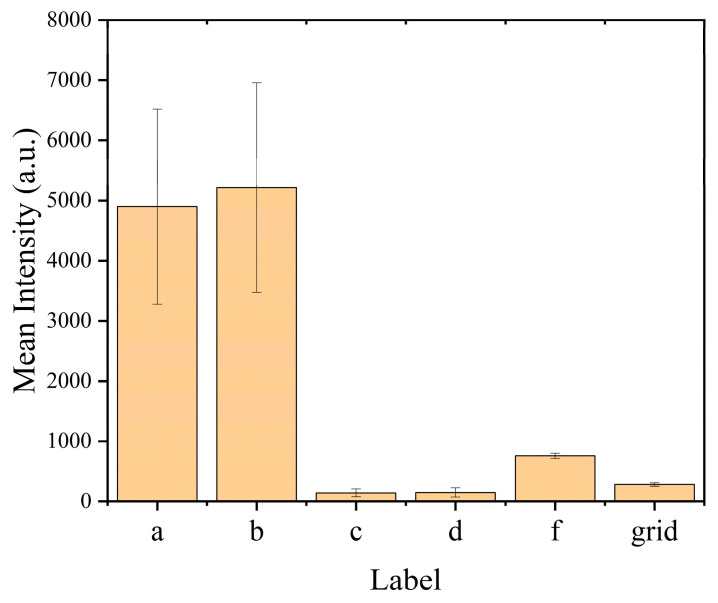
Fluorescence intensity of different cells and blank grid. The labels a, b, c, d and f correspond to the letter labels in Figure 4. The label grid indicates the autofluorescence of blank grid as a control. The fluorescence intensity of different cells was the net value after deducting the fluorescence of the blank grid. The number of pixels for calculating the mean intensity was, respectively, 2162, 1648, 1331, 1341, 2801 and 14,022.

**Figure 6 micromachines-14-00326-f006:**
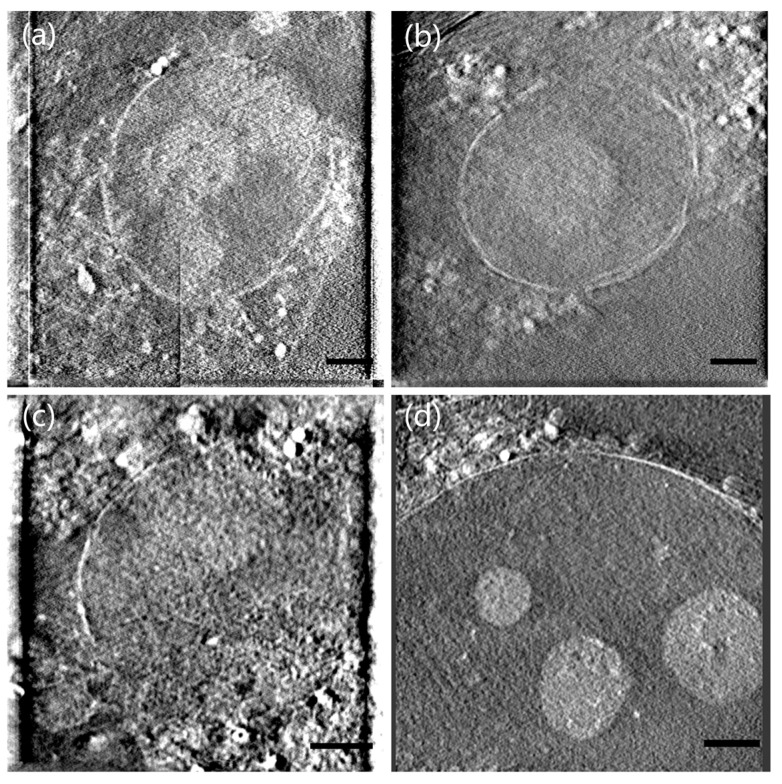
Reconstructed slices of HeLa cells. (**a**,**b**) slices of two apoptotic cells in Figure 4a; (**c**) slices of necrotic cell in Figure 4d; (**d**) slices of normal cell in Figure 4e. Scale bar 2 μm.

**Table 1 micromachines-14-00326-t001:** Criterion for determining cell status on-line.

Status	On-Line Fluorescence (Biochemical Information)	Soft X-ray Mosaic (Structural Information)
**Normal**	Faint signal	Intact structure
**Apoptosis**	Intense signal indicates caspase 3/7 activated	Cell shrinkage and Intact cell membrane
**Necrosis**	Faint signal	Cell swelling and Cytoplasm released

## Data Availability

The data in the manuscript can be obtained from the corresponding author.

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
