# Peer review of "On-Line Fluorescence Microscopy for Identification and Imaging of Apoptotic Cell with Synchrotron-Based Soft X-ray Tomography"

_micromachines, 2023, doi:10.3390/mi14020326_

Round 1

Reviewer 1 Report

The paper by Zhang et al. addresses the use of “online fluorescence microscopy” combined with SXT for the imaging of apoptotic cells in a synchrotron facility. The idea of this method is to reduce the time from finding a cell of interest for imaging from both modalities. It’s going to be useful if the method is well-designed for correlating fluorescence signals to SXT reconstructions. However, the writing and some other control experiments need to be addressed for consideration for publish.

(1)   Abstract part needs to be more concise to highlight the usage of online FM with SXT does provide advantages over other off-line FM techniques. There are some minor parts that raised my concerns.

a.     Line 9: to obtain ultra-structure images

b.     Line10: SXT can also be used for the quantitative study to calculate precise linear absorption coefficient (LAC) if a full-angle stage is used to acquire 180- or 360-degree projections

c.    Line17: the online FM is taking away some valuable beamtime. The authors need to modify the wording by not focusing on the beamtime but focus on “taking the useful data without tedious post-processing”

(2)   The introduction needs to be modified to highlight the advantages of using online FM over offline FM to save time to find an appropriate cell for SXT imaging. Especially, when the authors mentioned using super-resolution tools in the future for this end-station, I was wondering how complicated the processes will be to align the optics. How the working distance of the objective lens will hinder the rotation of the 2D grid sample stage.

a.     Line27: Full-angle SXT using capillary holder provides not only morphological information but also precisely quantitates the LAC

b.     Line30: It’d be helpful that the authors to highlight the issues for the ex-situ experiments from ref [9-15].

c.     Line48: The calculation of LAC from missing-wedge reconstructions needs to be further addressed, especially since the authors only take ~130-degree projections.

(3)   In Fig 4., the authors claimed there are cells treated with and without 0.5 μM staurosporine for 4 hours, where the untreated cell showed 20% of the fluorescence intensity of the apoptotic cells. It’s obvious to me that a solid control experiment (on the grid or just glass substrate) is needed to set up a baseline to distinguish the intensity from apoptotic and normal cells.

(4)   In Fig 5., the “N” is missing. What are the sample sizes for each condition for calculating the mean intensity?

(5)   Line 187, It’d be nice if the authors can address how the LACs were calculated. The unit of the LAC is missing, and how do those calculated LACs compare with other references?

(6)   Fig6., It’d be nice if the author can show a fully reconstructed “normal” cell to reveal the intact nuclear membrane for the comparison with “apoptotic cell”

Reviewer 2 Report

This manuscript reported a method based on the combination of synchrotron based soft X-ray tomography (SXT) and the on-line fluorescence microscopy, which facilitates the high-resolution imaging of apoptotic cells. However, some minor revisions are needed for further development of the manuscript. In my opinion, it is suitable for publication in “MDPI Micromachines”. There are minor comments on the manuscript listed below:

1. There is a scope for the more organized literature in introduction section. For example, a brief existing literature is needed using this combination (SXT and on-line fluorescence microscopy) in various cell organelles. Also, authors should focus on the recent literature (year: 2020-2022) regarding this.               

2. In section 2.1, there is a missing information on different algorithms such as FBP, SIRT and SART etc. Authors should abbreviate these terms and can explain the significance of these algorithms in the imaging techniques.

3. Authors should focus on the English grammar throughout the manuscript. For example, the sentence in the line 167, “the nucleolus in the nucleus are clearly visible”. But it should be “is” instead of “are”.

4. Authors should use abbreviations at first time in the manuscript. Further, use abbreviations itself instead of full forms. For instance, the “soft X-ray tomography” in the line 203.   

Round 2

Reviewer 1 Report

The questions were all nicely answered. I'd like to recommend this for publication.